Journal of Data-centric Machine Learning Research (2026)          Submitted 9/25; Revised 05/26; Published 05/26

# CleanPatrick: A Benchmark for Image Data Cleaning

**Fabian Gröger**[*1,2], **Simone Lionetti**[2], **Philippe Gottfrois**[1],
**Alvaro Gonzalez-Jimenez**[2,3], **Ludovic Amruthalingam**[2],
**Elisabeth V. Goessinger**[3], **Hanna Lindemann**[3], **Marie Bargiela**[3],
**Marie Hofbauer**[3], **Omar Badri**[5], **Philipp Tschandl**[6], **Arash Koochek**[7],
**Matthew Groh**[4], **Alexander A. Navarini**[†1,3], **Marc Pouly**[†2]

[1] *University of Basel*    [2] *Lucerne University of Applied Sciences and Arts*
[3] *University Hospital of Basel*    [4] *Northwestern University*
[5] *Northeast Dermatology Associates*    [6] *Medical University of Vienna*    [7] *Banner Health*

**Reviewed on OpenReview:** *https://openreview.net/forum?id=e2q3zXxm9v*

**Editor:** Andreas Kirsch

## Abstract

Robust machine learning depends on clean data, yet current image data cleaning benchmarks rely on synthetic noise or narrow human studies, limiting comparison and real-world relevance. We introduce CleanPatrick, the first large-scale benchmark for data cleaning in the image domain, built upon the publicly available Fitzpatrick17k dermatology dataset. We collect 496,377 binary annotations from 933 medical crowd workers, identify off-topic samples (4%), near-duplicates (21%), and label errors (32%), and employ an aggregation model inspired by item-response theory followed by expert review to derive high-quality ground truth. CleanPatrick formalizes issue detection as a ranking task and employs standard ranking metrics that mirror real audit workflows. We benchmark classical anomaly detectors, perceptual hashing, SSIM, Confident Learning, NoiseRank, FINE, BHN, and Self-Clean. On CleanPatrick, self-supervised representations excel at near-duplicate detection, classical methods achieve competitive off-topic detection under constrained review budgets, and detecting implausible labels under conservative human judgment remains challenging for fine-grained medical classification. By releasing both the dataset and the evaluation framework, CleanPatrick enables a systematic comparison of image-cleaning strategies.

**Benchmark:** github.com/Digital-Dermatology/CleanPatrick
**Dataset:** huggingface.co/datasets/Digital-Dermatology/CleanPatrick
**Keywords:** data quality issues, data cleaning, data-centric AI, benchmark

## 1 Introduction

The quality of training data is a cornerstone of effective machine learning (ML), with recent trends increasingly emphasizing data-centric approaches to boost model performance (Oquab et al., 2024). The quality of evaluation data is equally crucial, as contamination directly impacts how progress in the field is measured and the conclusions drawn from

---

*. Correspondence: fabian.groeger@unibas.ch

†. Joint last authorship

it (Northcutt et al., 2021b; Gröger et al., 2024). Currently, the evaluation of cleaning strategies, which could be used to resolve such issues, heavily relies on synthetic corruption of assumed to be clean datasets, *e.g.*, by artificially introducing noise or mislabeling samples (Northcutt et al., 2021b,a; Gröger et al., 2024). Although these controlled, synthetic setups offer repeatability, they often lack standardization since different studies adopt varied corruption protocols, making it challenging to compare results directly and benchmark progress across the literature. Furthermore, it is unclear how much a synthetic benchmark can mimic the nuances of real-world noise instead of solely favoring the authors' methods.

Several recent works have extended beyond synthetic methods to evaluate cleaning strategies on real-world contamination (Sharma et al., 2020; Northcutt et al., 2021b,a; Gröger et al., 2024). However, such evaluations tend to fall short in scope. Typically, these approaches repurpose annotations initially collected for other tasks (*e.g.*, multi-annotator assignments), while others rely on limited-sample human evaluations. Consequently, despite these efforts, a research gap remains in establishing a robust, universally applicable benchmark for data cleaning in the image domain, and thus a clear understanding of how well these approaches perform outside of their original setting.

In contrast to the general image domain, comprehensive benchmarks exist for cleaning structured data. For example, Abdelaal et al. (2023) introduced REIN, a benchmark framework for data cleaning in ML pipelines, while Li et al. (2021) explored the impact of data cleaning on classification performance in their CleanML study. These initiatives have not only standardized evaluation methods but have also driven rapid progress by providing clear, comparable metrics (Abdelaal et al., 2023; Li et al., 2021). Similar endeavors in data-centric AI, exemplified by benchmarks such as DCBench (Cui et al., 2022) and DataPerf (Mazumder et al., 2022), highlight the potential of well-defined benchmarks in driving advancements across diverse ML tasks.

Addressing these limitations for unstructured data, we introduce CleanPatrick, the first dedicated benchmark for data cleaning in the image domain featuring exhaustive annotation for three data quality issues. CleanPatrick originates from the Fitzpatrick17k dataset (Groh et al., 2021), a collection of 16,577 dermatological disease images collected from online dermatology lexicons. While built on a medical dermatology dataset, it is deliberately designed to serve as a challenging "stress test" for data cleaning algorithms. The domain's inherent complexities, such as fine-grained classes with subtle inter-class differences, a naturally long-tailed label distribution, and the importance of textural details, provide a realistic proxy for the nuanced, real-world data corruption that synthetically altered benchmarks lack. The data quality issues we address (*i.e.*, off-topic samples, near-duplicates, label errors) are universal. Thus by situating them in this difficult domain, CleanPatrick enables a robust evaluation of a method's true capabilities. For this benchmark, the original images were repurposed and comprehensively annotated for data quality issues. Specifically, the dataset was reviewed by medical crowd workers who identified off-topic samples, near duplicates, and label errors, following terminology established by recent works in data-centric ML (Gröger et al., 2024). To achieve meaningful results for near duplicates, we selected pairs of samples based on a carefully engineered iterative procedure that requires at most as many annotations as the size of the dataset. Across the three issue types, we collected 496,377 annotations from 933 unique annotators. The resulting data-cleaning benchmark provides a realistic representation of contamination as it occurs in practice, especially when

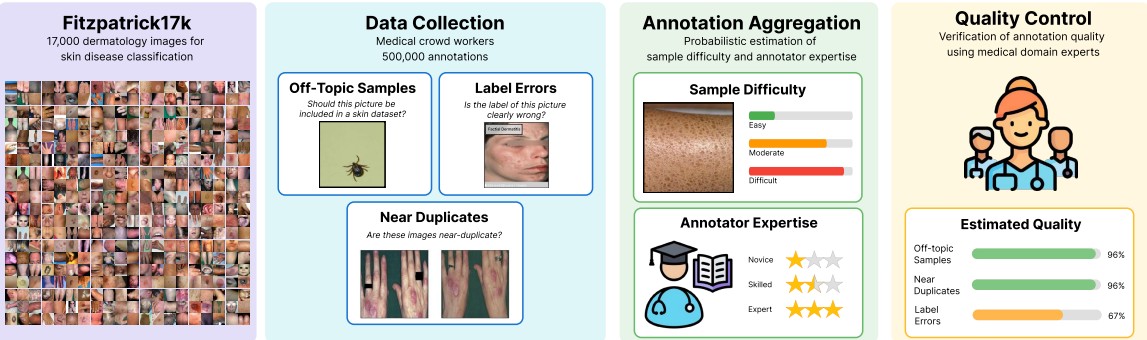

Figure 1: Process of acquiring and curating the CleanPatrick benchmark. We began by collecting annotations from medical crowd workers for three types of data-quality issues. This was followed by a probabilistic estimation of sample quality and annotator expertise, used to aggregate the collected annotations. Finally, a group of medical domain experts judged the quality of a subsample of the dataset.

obtained through a semi-automatic procedure, moving beyond the constraints of synthetic evaluations.

By standardizing the evaluation procedure with CleanPatrick, we aim to facilitate a fair and detailed comparison of different cleaning strategies. Our benchmark not only builds on the lessons learned from structured data cleaning but also addresses the unique challenges inherent to image data, where real-world contamination is often more nuanced and complex than what synthetic corruptions can capture. Ultimately, CleanPatrick lays the foundation for future innovations in curation approaches by providing a comprehensive resource that bridges the gap between traditional synthetic evaluations and the demands of real-world applications.

When evaluating existing methods for detecting different types of data quality issues, we found that while near duplicates are relatively easy for human experts to detect, as reflected in the high inter-annotator agreement, they are challenging for existing methods, especially when they result from a part-whole relationship. Off-topic samples are difficult for current approaches to detect, but they are relatively easy for human experts to identify when given precise instructions. Label errors are both difficult for human experts to detect, as reflected by the lower inter-annotator agreement compared to the other issue types, and for current approaches, likely due to the challenging nature of the dataset. Overall, while current approaches already achieve promising results, there remains substantial room for improvement, particularly in detecting context-dependent data-quality issues, such as those present in uncurated medical imaging datasets.

In summary, the main contributions are: (1) The release of the first data cleaning benchmark for images obtained from 496,377 annotations from medical crowd workers and verified by medical experts. (2) The outline of a standardized procedure for evaluating data cleaning methods. (3) The comparison of existing approaches for detecting diverse data quality issues and an analysis of their failure cases.

## 2 Related work

Traditionally, the evaluation of data cleaning methods in computer vision has been carried out by introducing synthetic corruptions into believed-to-be clean datasets. For instance, frameworks such as SelfClean (Gröger et al., 2024) and Confident Learning (Northcutt et al., 2021b,a) simulate realistic noise, mislabeled samples, or other forms of data contamination using artificial perturbations. These evaluation strategies are beneficial since they can be programmatically generated for any dataset and produce many variations. However, they inherently rely on assumptions about the nature of contamination that may not fully capture the complexity of real-world data errors.

In the domain of dermatology, several studies have reported challenges arising from real contamination in clinical image datasets. Analyses of widely used dermatological image datasets have uncovered significant issues, including data leakage between training and testing splits, near-duplicate images, off-topic samples, and non-standardized or erroneous diagnostic annotations (Gröger et al., 2023; Abhishek et al., 2025; Gröger et al., 2025). These studies, though valuable, often rely on limited-scale human evaluation or the indirect reuse of annotations, and instead aim to release new and improved versions of established benchmarks rather than evaluating the methods used during the procedure. This illustrates a fundamental problem: since these studies rely on existing tools or heuristics to speed up data cleaning, using their annotations for evaluation makes a true, unbiased comparison virtually impossible. In this paper, we do not use any tools or methods for performing the data selection, but instead rely on exhaustive annotation and a large pool of medical crowd workers to obtain unbiased annotations.

While the image domain has lagged behind in standardized evaluation procedures, the field of structured data cleaning has seen considerable progress in recent years. Benchmarks such as REIN by Abdelaal et al. (2023) and CleanML by Li et al. (2021) offer comprehensive frameworks and quantitative metrics for comparing cleaning methods. Similarly, tools integrated within data-centric AI platforms, such as DCBench (Cui et al., 2022) and DataPerf (Mazumder et al., 2022), provide a systematic evaluation environment that has spurred rapid advancements in data quality for structured data. These benchmarks have not only provided a level playing field for method comparison but have also driven progress by clearly highlighting the impact of data quality on downstream model performance. However, extending these insights to the unstructured image domain remains challenging due to the inherently different nature of visual data and the difficulty of designing detailed instructions for annotation tasks.

For label error detection, the AQuA benchmark (Goswami et al., 2023) injects seven synthetic noise patterns (uniform, asymmetric, class-dependent, instance-dependent, dissenting label, dissenting worker, and crowd majority) across 14 vision, text, and tabular datasets, creating a fully controlled testbed for cleaning algorithms. Human re-annotation efforts include CIFAR-10H (Peterson et al., 2019), which provides 511,400 crowd labels for the CIFAR-10 test set, yielding a soft ground-truth distribution widely used to evaluate label quality. For unsupervised near-duplicate detection, Morra and Lamberti (2019) released a benchmark with verified duplicate pairs that cover common web transformations. For off-topic or out-of-distribution images, robustness benchmarks such as ImageNet-O, featur-

Table 1: Performance of label error detection methods on CleanPatrick and CIFAR-10N. Values (in %) represent the mean with 95% confidence intervals obtained via bootstrapping.

| Method | CleanPatrick ($p^+ = 32.3$) | | CIFAR-10N ($p^+ = 9.1$) | |
| | AUROC | AP | AUROC | AP |
| --- | --- | --- | --- | --- |
| CLearning | 48.1 [47.1, 49.0] | 31.7 [30.8, 32.8] | 55.5 [54.6, 56.4] | 11.0 [10.6, 11.6] |
| NoiseRank | 48.2 [47.2, 49.1] | 31.1 [30.2, 32.1] | 57.3 [56.5, 58.2] | 11.1 [10.7, 11.6] |
| SelfClean | 58.3 [57.3, 59.2] | 38.7 [37.6, 39.9] | 61.1 [60.1, 62.0] | 13.4 [12.9, 14.1] |

ing 2,000 categories absent from ImageNet-1k, measure false-positive rates when irrelevant classes appear at test time (Hendrycks et al., 2021).

While these specialized resources have driven progress for their error types, they assess methods in isolation and often on general-domain imagery. Furthermore, none of the existing benchmarks directly obtains labels for data quality issues. Obtaining labels by re-labeling rather than asking annotators to identify quality issues yields markedly different outcomes. Instead, the outlined benchmark corresponds to human assessment of the issues themselves. CleanPatrick advances the field by providing expert-verified annotations for all three major data-quality problems (label errors, near duplicates, and off-topic images) within a single, clinically realistic dataset, enabling holistic evaluation and direct cross-method comparison.

CleanPatrick addresses an overlapping objective compared to standard learning with noise. Weak supervision and noisy-label methods typically aim to *train robust models* despite label noise or to *recover true latent labels* using assumptions about the noise process (*e.g.*, class-conditional or instance-dependent noise). In contrast, CleanPatrick targets *audit prioritization* to avoid data poisoning, by identifying the labels that a human inspector would most easily identify as incorrect. As such, CleanPatrick provides a distinct evaluation axis that complements existing noisy-label benchmarks, testing whether methods can prioritize the most unambiguous errors for efficient human review.

To better position our work and assess the generalizability of a domain-specific benchmark, we conduct a direct comparison with CIFAR-10N, a key benchmark for real-world label noise generated from human annotations. While CIFAR-10N is a crucial resource, CleanPatrick is designed as a complementary, more challenging benchmark that addresses a wider array of data-quality issues, including not only label errors but also near-duplicate and off-topic samples. Our methodology also differs. CleanPatrick provides a direct catalog of verified issues within the original dataset, whereas CIFAR-10N provides new annotations for CIFAR-10 and uses the original labels as the noisy set for comparison. To demonstrate that insights from our benchmark can transfer to other domains, we evaluated the same suite of label error detection methods on both datasets. The results in table 1 show that while absolute performance differs due to the varying difficulty and class imbalance ($p_+$), the relative ranking of methods is consistent, with SelfClean outperforming the others on both benchmarks. This suggests that CleanPatrick can serve as a valuable, more challenging counterpart to general-domain benchmarks, thereby driving the development of more sophisticated and broadly applicable data cleaning algorithms.

## 3 The CleanPatrick benchmark

This section details how we transform the Fitzpatrick17k collection into CleanPatrick, a rigorously annotated benchmark for data cleaning research. We proceed as follows: Section 3.1 revisits the provenance and characteristics of Fitzpatrick17k, Section 3.2 describes the large-scale annotation campaign with medical crowd-workers, Section 3.3 explains how we aggregate the (noisy) votes with item-response theory, Section 3.4 reports the independent quality-control performed by medical domain experts, and finally, Section 3.5 formalizes the three tasks and their evaluation metrics.

### 3.1 Fitzpatrick17k

Fitzpatrick17k is a public collection of 16,577 clinical photographs covering 114 distinct skin disease diagnoses and labeled with Fitzpatrick skin types (I–VI) (Groh et al., 2021). Images were obtained from two open-access dermatology atlases, DermNet (12,672 images) and Atlas Dermatológico (3,905 images), and are released under a CC BY-NC-SA 3.0 license. Compared with other dermatology datasets, it is relatively large, more diverse across both disease spectrum and skin tone distribution, and can be considered weakly supervised, as labels were extracted from atlas captions rather than from structured clinical metadata. The original taxonomy groups diseases into 114 classes, although the dataset features two additional, coarser-grained levels that were obtained during post-processing. For CleanPatrick, we retain the finest granularity to preserve compatibility with prior work while keeping the data as close as possible to the originally obtained collection. The dataset has been subject to thorough analysis (Daneshjou et al., 2022; Gröger et al., 2024; Yan et al., 2024; Abhishek et al., 2025), in which previous studies estimated between 16%–30% problematic images. This real-world noise and challenges of a medical dermatological dataset (*i.e.*, diverse skin tones and unbalanced classes) motivated our decision to start with Fitzpatrick17k.

### 3.2 Annotation process

We decomposed the annotation process of the data-quality issues into three independent tasks according to their type, *i.e.*, off-topic samples, near duplicates, and label errors, and deployed each as a separate labeling task on Centaur Labs[1], a platform that screens contributors for medical knowledge and thus has access to a large collective of medical crowd workers. Precise instructions, including examples, were formulated in collaboration with dermatologists and annotation specialists from Centaur Labs (see Appendix D). Additionally, we collected a few hundred gold-standard samples for each labeling task and used them to obtain immediate feedback on the accuracy of the collection, to educate annotators, and to filter out inattentive or adversarial raters. These gold standards were obtained from unanimous agreement of domain experts for off-topic samples and near duplicates, and from three board-certified dermatologists for label errors.

The following paragraphs summarize the labeling task description for each issue type, as provided to both crowd workers and expert annotators. Their exact formulations, including screenshots of the labeling platform, are provided in Appendix D.

---

1. `https://www.centaurlabs.com/`, accessed on 26th of September 2025.

**Off-topic samples.** The task of the annotators was to determine if a picture was included in a dataset of human skin lesions by mistake and is, therefore, off-topic in the dataset's context. For each sample, we asked the annotators *should this picture be included in a dataset of skin condition images?* where they were expected to answer with *yes* if the image correctly showed a skin condition and *no* if the image did not meet the criteria for inclusion in the dataset. Reasons to not include the image in the dataset were, for example, that the image is from a different modality (*e.g.*, an X-ray or a PowerPoint slide with mostly text) or that the image did not focus on a skin condition (*e.g.*, it shows a fully clothed patient without any visible skin disease). Pictures should be included if they are photos of human skin diseases. In the instructions, we also provided examples of typical images from other skin condition datasets and some examples of images that were likely included by mistake.

**Near duplicates.** The task of the annotators was to determine if two pictures, shown side by side, were near duplicates. We thus asked the annotators *are these images near-duplicates?* and asked them to answer *no* if the images were not related and *yes* if these images are transformations of one another (*e.g.*, rotations, flips, image edits), or nearly identical because they were taken within seconds of each other, or some other reason which created a relationship among the samples. In the instructions, we also provided examples of both near-duplicate and non-duplicate pairs.

To avoid the prohibitive $\mathcal{O}(N^2)$ effort of exhaustively judging all $\binom{N}{2}$ image pairs in a dataset with $N$ samples, we introduce a *fast-duplicates* procedure (see Appendix F) which speeds up the annotation process by relying on batch-wise annotation. Specifically, each image $x_i$ is embedded with an encoder that was pre-trained on ImageNet with self-supervision, presently DINO (Caron et al., 2021). Its nearest neighbor $n(x_i)$ is then retrieved, and only the at most $N$ unordered pairs $\{x_i, n(x_i)\}$ are sent to annotation by crowd workers. This process is then repeated after the annotation is finished for the current batch of positively annotated (*i.e.*, near duplicate) samples. Under the *fast-cleaning* assumption that every near duplicate of $x_i$ is closer to $x_i$ than any non-duplicate, this strategy discovers all duplicate cliques in at most $\lfloor \log_2 K \rfloor + 1$ rounds, with $K$ being the size of the largest clique, while requiring no more than $2N$ pairwise judgments in total (see Lemma 1). In the case at hand, the procedure stopped after nine batches with duplicates and one with all negative responses, and the batch size dropped exponentially as expected (see Figure 10 in the Appendix).

Despite independent human verification, candidate retrieval based on ImageNet DINO embeddings may bias evaluation in favor of similar approaches. We audit the corresponding effect with a sample of candidate pairs from other methods in Appendix H. Pixel-based methods (pHash, SSIM) yield no additional confirmed duplicates in the sample, so the Wilson 95% confidence interval gives a maximum prevalence of 2.5% in their top-5,000 candidates. Moreover, SelfClean's AUROC is preserved within 0.1% under re-evaluation with 17 additional near-duplicate pairs identified using supervised ImageNet representations. The audit therefore supports CleanPatrick as a reliable testbed for near-duplicate detection.

**Label errors.** The task of the annotators was to determine if a picture was wrongly annotated. We thus showed a single picture along with its originally assigned diagnosis and asked the annotators, *is the label of this picture clearly wrong?*. The annotators were expected to answer *yes* if the diagnosis was clearly wrong for the given image and *no* if

the diagnosis was not a clear label error. In the instructions, we also provided examples of clearly incorrect and correctly annotated samples. We explicitly stated that if the diagnosis for a skin lesion is likely incorrect but could be correct under special circumstances, the annotation is not clearly wrong and should not be considered a label error, because the goal was to identify errors rather than unlikely or ambiguous cases. Furthermore, because some diagnoses can be rare or difficult to assess, we recommended that experts consult online dermatological atlases, such as DermWeb, when unsure about the condition.

Using this procedure, we collected a total of 496,377 binary votes from 933 medical crowd workers, with some annotators contributing as many as 15,630 annotations and others as few as 1. For each sample, we collected an average of 10 votes, with some samples having as many as 225 and others only 1. The raw annotation data, *i.e.*, the vote of each unique annotator, can be found in the released dataset. Appendix E contains a detailed analysis of the annotations, while Appendix J estimates the impact of samples with low annotation count on the benchmark.

### 3.3 Label aggregation

To best leverage the wealth of annotations from medical crowd workers, we need to consider that annotators differ widely in skill and commitment. Some will be novices in dermatology, whereas others are experts, and since we have limited influence on recruitment, we should consider adversaries, *i.e.*, annotators intentionally not solving the task. Thus, we utilize ideas from item response theory (IRT), where one can model the skill of an annotator and the difficulty of a sample, instead of assuming that all annotators and samples are equal, as typically done with majority voting. The following section describes the IRT model employed to probabilistically estimate the difficulty of a sample and the ability of an annotator, which are then used to obtain the final labels.

Let $\mathcal{Y} = \{(a, i, y_{a,i})\}$ be the set of $Y$ noisy binary annotations collected from the medical crowd-workers through the process outlined above, where each tuple records *who* (annotator $a$ of total $A$) labeled *what* (item $i$ of total $I$) and the observed binary response $y_{a,i} \in \{0, 1\}$. Because most annotators label only a fraction of the items, the resulting observation matrix is sparse.

We adapt the *Generative Model of Labels, Abilities, and Difficulties* (GLAD) (Whitehill et al., 2009) to our setting. GLAD assumes that the probability of a correct annotation depends multiplicatively on annotator ability and item difficulty:

$$\Pr(y_{a,i} = 1 \mid c_a, b_i) = \sigma(c_a b_i), \qquad \sigma(x) = \frac{1}{1 + e^{-x}},$$

where $c_a$ is the expertise or ability of annotator $a$ and $b_i \in \mathbb{R}$ captures the difficulty of item $i$. The generative process is modeled by

$$y_{a,i} \mid c_a, b_i \sim \text{Bernoulli}\left(\sigma(c_a b_i)\right).$$

We make two key modifications to the original formulation. First, we drop the exponential parametrization of $b_i$ such that $b_i \in \mathbb{R}$, allowing positive and negative values to encode the positive and negative latent classes, respectively. This unifies "difficulty" and "class orientation" in a single parameter, where small $|b_i|$ means difficult, and the sign of

$b_i$ reveals the latent class. Second, we choose the priors

$$c_a \sim \mathcal{N}(0, 1), \qquad b_i \sim \mathcal{N}(0, \sigma_b^2),$$

with $\sigma_b = 10^3$ giving a *vague* prior for difficulty, and the starting abilities centered around zero. Compared to the original $c_a \sim \mathcal{N}(1, 1)$, this choice reflects a more pessimistic prior due to low annotator control, since zero corresponds to chance-level performance, positive values indicate expertise, and negative values a performance below chance that possibly indicates adversarial behavior.

**Variational inference.** Since exact posterior inference is intractable, we use stochastic variational inference (SVI) as implemented in PYRO (Bingham et al., 2019). The mean-field variational family factorises over latent variables:

$$q(c_1, \ldots, c_A, b_1, \ldots, b_I) = \prod_{a=1}^{A} \mathcal{N}\big(c_a \mid \mu_{c_a}, \sigma_{c_a}^2\big) \prod_{i=1}^{I} \mathcal{N}\big(b_i \mid \mu_{b_i}, \sigma_{b_i}^2\big).$$

We optimize the evidence lower bound (ELBO) with Adam (Kingma and Ba, 2015) (learning rate 0.1) for 10,000 steps, which is sufficient for convergence on all splits.

**Predictive aggregation via the $b_i$ distribution.** Under the assumption that most annotators are not adversarial, the sign of $b_i$ encodes the most likely class of sample $i$. To estimate the probability of a data point to belong to the positive class, we draw $M = 1,000$ samples $\{b_i^{(m)}\}_{m=1}^{M}$ from each $b_i$'s variational posterior and compute

$$\bar{p}_i = \frac{1}{M} \sum_{m=1}^{M} \mathbb{I}[b_i^{(m)} > 0].$$

The distribution of the $\bar{p}_i$ for the different issue types is given in Figure 6 of the appendix. Our aggregation model is aware of uncertainties and estimates that 1.3% of near duplicates, 8.4% of label errors, and 1.3% of off-topic samples have a wrongly assigned label.

### 3.4 Quality control

To estimate the quality of annotations from medical crowd workers, we recruited three medical domain experts with more than five years of experience as practicing dermatologists, after they completed their medical examinations. Since a full new annotation was not possible due to resource limitations, we use a quantile-stratified random sampling scheme to ensure uniform coverage of the entire probability range $\bar{p}_i$. Specifically, we split the probabilities into 20 bins and randomly selected 20 samples from each bin, resulting in a total of 400 samples per data quality issue type. The medical experts followed the same protocol as outlined in Section 3.2 and instructions as the medical crowd workers.

We computed inter-annotator reliability for the three issue types in terms of Krippendorff's $\alpha$ and Cohen's $\kappa$. Krippendorff's $\alpha$, computed over all three raters, shows excellent agreement for near-duplicate annotations ($\alpha \approx 0.91 \pm 0.03$). It falls to a lower agreement level for off-topic samples ($\alpha \approx 0.60$ with a wide 95% CI spanning 0.20–0.85, caused by the high imbalance between problematic and non-problematic samples) and drops further

for label errors ($\alpha \approx 0.42 \pm 0.06$). Consistent with Krippendorff's $\alpha$, Cohen's $\kappa$ values are near 0.90 for near duplicates regardless of the annotator pair, whereas they fluctuate much more for off-topic samples and exhibit very large confidence intervals for some annotator pairs. Agreement on label errors is the weakest, at the boundary between fair and moderate agreement on the conventional Landis–Koch scale (Landis and Koch, 1977), and below Krippendorff's reliability threshold of 0.667. In summary, the agreement shows that verification is consistent among the experts for off-topic samples and near duplicates, while the lower agreement on label errors propagates some uncertainty to the LE ground truth (Lionetti et al., 2025). Appendices G and I quantify the impact of this expert-level uncertainty on benchmark conclusions, demonstrating that the relative ranking of label-error detection methods is robust. This is still not trivial for a task as complex as label error detection, which other studies have found to be substantially difficult (Ribeiro et al., 2019; Krefting et al., 2024; Gröger et al., 2023). Figure 11 in the appendix summarizes the results for the inter-annotator reliability.

In order to estimate the quality of the crowd annotations, we aggregated the expert annotations using majority voting and compared them. Experts and crowd workers agree on 96% of the verified images for off-topic samples and near duplicates, and on 67% of label errors. This further demonstrates that a carefully designed protocol is helpful to achieve high-quality annotations at scale with the help of medical crowd workers, even for complex tasks.

Additionally, we use expert annotations to estimate the threshold for obtaining the final labels for each data quality issue separately, rather than naively choosing a fixed value of 0.5. For this, we use the aggregated expert annotation and the bins we defined above, and check the distribution of labels for each one. We choose the threshold $t$ at the bin where the distribution of positive labels starts to increase and select the threshold as the average probability of that bin. The final label is then obtained using this estimated threshold, which is different for each issue type:

$$\hat{y}_i = \mathbb{I}[\bar{p}_i \geq t].$$

### 3.5 Evaluation tasks

We cast the detection of data-quality issues as a *ranking* problem. Rather than producing a binary keep/discard decision, each method must assign a real–valued score that reflects how strongly an example (or example pair) is suspected to be a data quality issue. Prior work has shown that practitioners subsequently inspect items in descending score order, making ranking the most faithful abstraction of real-world use (Gröger et al., 2024; Gröger et al., 2023).

**Tasks.** The benchmark comprises three evaluation tasks, one for each quality issue type:

1. **Off–topic sample detection.**
   *Input:* a single image $x_i$.
   *Output:* an anomaly score $s(x_i) \in \mathbb{R}_{\geq 0}$, where larger scores indicate a higher likelihood that the image does *not* depict a skin condition.
   *Positive criterion:* the image is off–topic as identified by the medical crowd workers.

2. **Near-duplicate detection.**
   *Input:* an unordered image pair $(x_i, x_j)$.
   *Output:* a similarity score $s(x_i, x_j) \in \mathbb{R}_{\geq 0}$ reflecting the confidence that the pair is a near duplicate.
   *Positive criterion:* the pair belongs to the same near-duplicate component identified. Note that we only compare the annotated samples, *i.e.*, we do not treat the unannotated pairs as negative to have a more reflective performance estimation of the methods.

3. **Label-error detection.**
   *Input:* an image $x_i$ together with its original diagnosis $y_i$.
   *Output:* a confidence score $s(x_i) \in \mathbb{R}_{\geq 0}$ that the assigned label $y_i$ is *incorrect*.
   *Positive criterion:* medical crowd workers judged the label to be "clearly wrong".

**Task semantics.** The three issue types in this benchmark are defined as follows:

- **Off-topic samples** are images that do not belong in a dataset of skin conditions (*e.g.*, different modalities, non-dermatological content, or lacking diagnostic value).

- **Near duplicates** are image pairs that are transformations of one another or depict the same skin lesion from potentially different viewpoints. These pairs are limited to candidates that are extracted iteratively as nearest neighbors under general self-supervised representations.

- **Label errors** specifically refer to *implausible annotations*, *i.e.*, cases where the assigned diagnosis is visually incompatible with the image content under conservative human judgment. This focus on unambiguous cases enables high inter-annotator agreement, providing a clean signal for method evaluation.

**Metrics.** Methods are evaluated using standard ranking metrics, such as P@k, R@k, area under the receiver operating characteristic curve (AUROC), and average precision (AP). AUROC and AP are the primary metrics, while P@k and R@k illustrate the practical trade-offs between effort and gain for review budgets of $k \in \{100, 500, 1000\}$ images. Additionally, we report the proportion of positive samples $p^+$, which corresponds to the baseline AP. To ensure statistical robustness, we report 95% confidence intervals for all key metrics, computed using 2,000 bootstrap samples.

## 4 Results

### 4.1 Data quality issues

Our extensive annotation process with medical crowd workers revealed numerous data quality issues in the Fitzpatrick17k dataset, as illustrated in Figure 2.

**Off-topic samples** constitute 613 images (4%) of the full dataset. These problematic samples fall into two main categories: *unrelated* content, including non-dermatological images such as laboratory equipment, diagrams, and completely unrelated photographs, and *low information content* images that, while potentially skin-related, lack sufficient clarity

or focus to be diagnostically meaningful. As illustrated in Figure 2, this explicitly includes severely blurred images, extreme close-ups with minimal context, or images where the skin condition is barely visible, thus capturing critical aspects of poor image quality.

**Near duplicates**   form a substantial portion of the dataset, with 3,556 instances (21%) out of the 15,306 annotated samples. These appear primarily as *thumbnails*, where identical images exist at different resolutions or with minor cropping differences, and *multiple viewpoints* of the same skin condition from slightly different angles or captured moments. This redundancy artificially inflates certain diagnostic categories and may introduce data leakage between the training and test splits.

**Label errors**   represent the most prevalent issue, affecting 5,346 images (32%) of the full dataset under our conservative annotation protocol. These visually implausible labels manifest as cases where the assigned diagnostic label *clearly contradicts* the visible condition, and *rare conditions* that were incorrectly classified, likely due to their uncommon presentation or similarity to more common conditions. The prevalence of visual mismatches that humans consider unambiguous provides a substantial and reliable test set for evaluating label-error detection methods.

The significant prevalence and diversity of these naturally occurring issues make the Fitzpatrick17k dataset an ideal foundation for a medical data cleaning benchmark. Unlike synthetic corruptions that artificially introduce noise following predetermined patterns, these issues represent authentic challenges that data cleaning algorithms must address in real-world applications. The distribution of issue types (4% off-topic, 21% near duplicates, 32% label errors) provides a comprehensive test bed that spans the spectrum of common data quality problems. This natural distribution is particularly valuable for benchmarking, as it reflects one example of contamination encountered in practice rather than artificially balanced scenarios. Furthermore, having ground truth for these issue types enables precise evaluation of detection algorithms across varying difficulty levels, from the relatively straightforward identification of off-topic samples to the more nuanced task of detecting label errors in specialized medical imagery. This comprehensive characterization of data quality issues establishes CleanPatrick as a robust, realistic benchmark for advancing data cleaning methodologies in the image domain.

## 4.2 Benchmark results

The detailed performance metrics for all evaluated methods are presented in Table 10 in the appendix. For visual clarity, Figure 3 summarizes these results by showing the performance of the best two methods for each issue type in terms of AUROC, AP, and precision/recall at review budgets $k = \{100, 500, 1000\}$. Below, we describe these results and discuss their implications for real-world data-cleaning workflows.

**Off-topic sample detection.**   Classical anomaly detectors, such as IForest, HBOS, and ECOD, achieve similar overall rankings (AUROC $\approx$ 0.76–0.77, AP $\approx$ 0.15–0.16). In contrast, SelfClean, a dedicated data cleaning strategy, attains an AUROC of 0.67 and AP of 0.14 but exhibits higher precision for the top 100 candidates (P@100 = 0.52) compared to the

---

2. Note that the categorization is solely used for visualization and not part of the released benchmark.

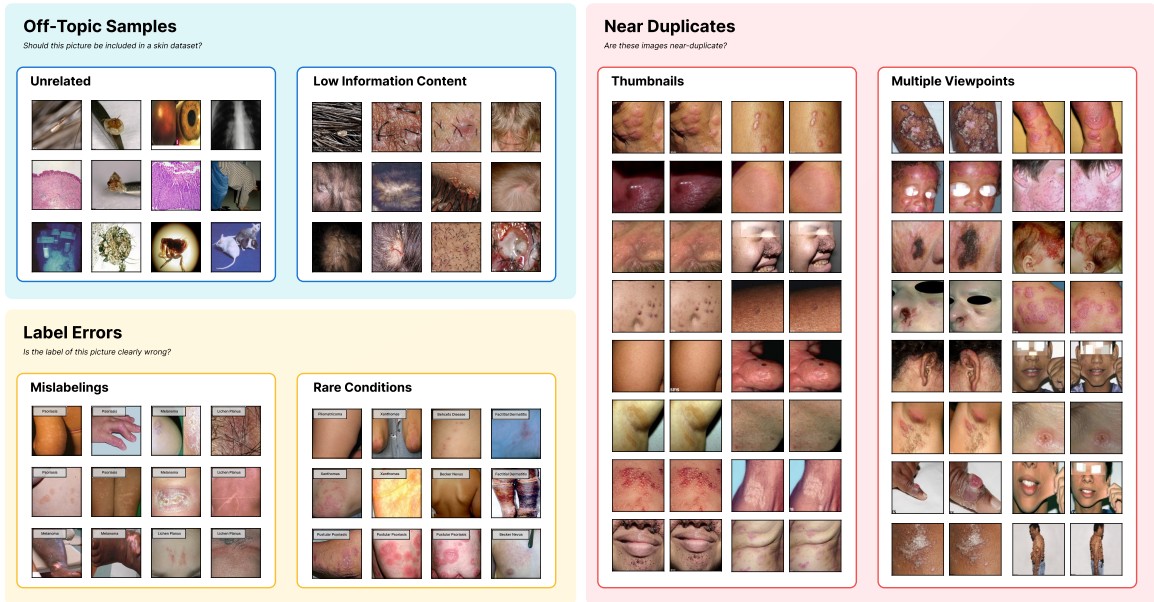

Figure 2: Examples of data quality issues identified in the Fitzpatrick17k dataset.[2] **Off-topic samples** include unrelated content (*e.g.*, laboratory equipment, diagrams) and images with insufficient diagnostic value. **Near duplicates** comprise identical images at different resolutions (thumbnails) and multiple photographs of the same condition from different angles. **Label errors** show both clear mislabelings and rare conditions that were incorrectly classified or assigned. These naturally occurring issues form the foundation of the CleanPatrick benchmark, providing a realistic test scenario for evaluating data cleaning algorithms across varying levels of detection difficulty.

other methods. This indicates that while SelfClean's global scores are less calibrated, its highest-confidence predictions are more reliable under limited review budgets compared to classical anomaly detectors.

**Near-duplicate detection.** Perceptual hashing and SSIM perform near chance (AUROC ≈ 0.50, AP marginally above 0.31), reflecting their difficulty in capturing the subtle duplicates present in CleanPatrick. SelfClean, by leveraging self-supervised embeddings, achieves an AUROC of 0.92 and an AP of 0.88, with P@100–P@1000 = 1.00. This gain highlights the effectiveness of representation learning in duplicate detection, consistent with previous findings (Fernandez et al., 2022; Gröger et al., 2024). To audit the retrieval mechanism for bias, we sampled and annotated 450 additional high-similarity pairs from pHash, SSIM, and an ImageNet-supervised ViT-T (Appendix H). Table 4 confirms that SelfClean's AUROC is preserved while pHash and SSIM lose 1–2 AUROC points. Of the 17 newly confirmed duplicates, SelfClean recovers 76.5% within a review budget of $k = 5{,}000$, compared to 17.6–35.3% for pixel methods (Table 5). The advantage of representation learning on near-duplicate detection is thus confirmed.

**Label error detection.** Detecting visually implausible labels in fine-grained medical imaging represents a challenging test case. FINE, NoiseRank, and Confident Learning all

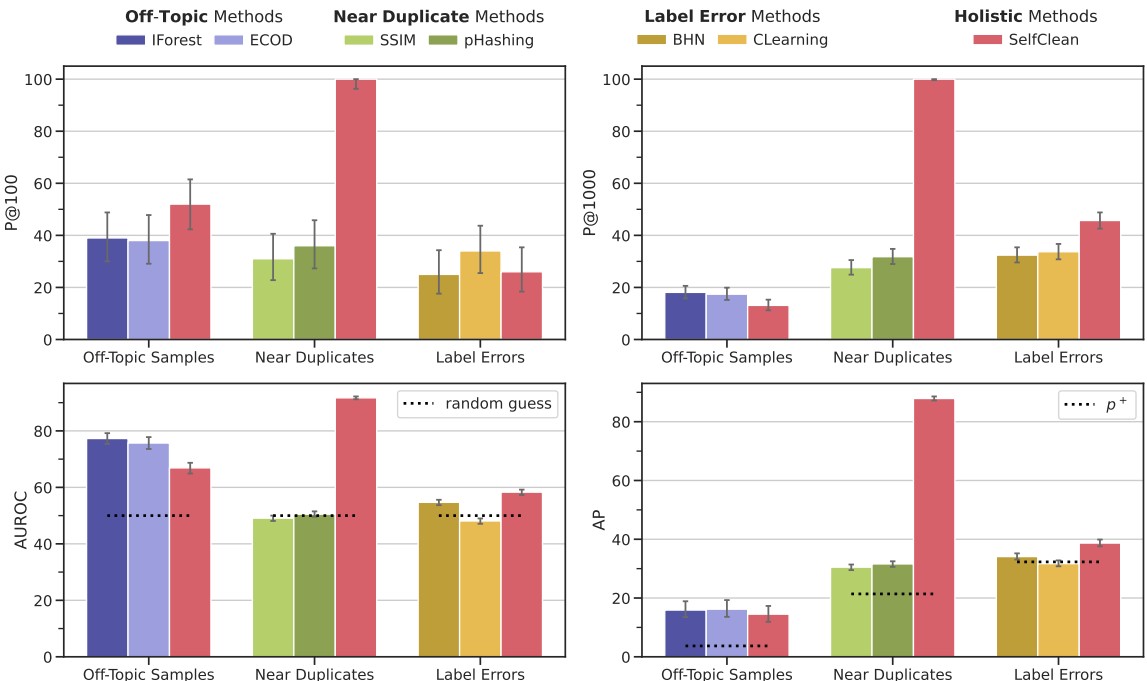

Figure 3: Performance of different data cleaning approaches (represented in colors) for the three quality issues investigated for different ranking metrics (P@100, P@1000, AUROC, and AP). Methods are categorized into data quality issue-specific ones and holistic methods that can detect multiple issues. Error bars represent 95% confidence intervals computed via bootstrapping. The dotted lines refer to the uninformed baseline, which randomly shuffles the ranking.

perform near or below the random baseline (AUROC $\leq$ 0.48, AP $\leq$ 0.32). BHN shows a modest improvement (AUROC: 0.55, AP: 0.34), while SelfClean achieves the best performance (AUROC: 0.58, AP: 0.39). Notably, even the best-performing methods achieve precision at small review budgets near or below the base rate (P@100 $\leq$ 0.35), indicating that none of the methods reliably prioritize label errors among the top-ranked predictions. These results reveal that CleanPatrick tests a distinct capability from standard noisy-label benchmarks: prioritizing *visually unambiguous* mismatches to prevent data poisoning and loss of trust, rather than modeling class-conditional noise distributions.

The difference between global ranking metrics and top-$k$ precision highlights a critical trade-off in data-cleaning methods, namely that methods optimized for AUROC or AP may not prioritize the most egregious errors when annotation budgets are constrained. Self-Clean's holistic, representation-based approach excels at surfacing high-confidence anomalies, particularly duplicates, making it well-suited for audits with budget constraints. However, its limitations in label-error detection imply that hybrid pipelines, which combine specialized, domain-aware detectors with self-supervised models, may yield better overall coverage. The persistent challenge of label noise, however, invites future research into integrating metadata, human-in-the-loop feedback, or multi-stage detection strategies.

## 5  Conclusion

In this work, we introduced CleanPatrick, the first benchmark for data cleaning in the image domain. Building on the publicly available Fitzpatrick17k dataset for skin disease classification, we collected 496,377 annotations from 933 medical crowd workers, which were further validated through expert review. This process helped identify data-quality issues of three types: off-topic samples, near duplicates, and label errors. We formalized each detection task as a ranking problem with standardized evaluation metrics (AUROC, AP, P@k, R@k) and provided clear protocols for annotation, aggregation via a model inspired by item-response theory, and expert-driven threshold selection. In extensive experiments, we found that, on this benchmark, near-duplicate detection benefits greatly from self-supervised representations, off-topic detection is addressed well by classical anomaly detectors, achieving higher top-k precision under limited review budgets, and detecting visually implausible labels under conservative human judgment remains challenging, with even the best methods achieving only modest improvements over chance. By releasing both the CleanPatrick dataset and an accompanying evaluation framework, we provide a realistic testbed that moves beyond synthetic corruptions and captures the nuanced, real-world contamination patterns encountered in medical imaging. By serving as a stress test with its fine-grained and complex nature, it pushes the boundaries of current methods and paves the way for developing more broadly applicable data cleaning algorithms.

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

## Appendix A. Limitations and Broader Impact

**Limitations.** Attention should be paid to the following limitations: (1) By presenting annotators with only each image's nearest neighbor rather than exhaustive, global pairwise comparisons, the assumption that all near-duplicates lie closer in embedding space than any non-duplicate may lead to incomplete discovery of near-duplicate groups. Appendix H bounds the resulting candidate-retrieval bias: pixel-based methods (pHash, SSIM) miss at most 2.5% of duplicates within their top-5,000 pairs (Wilson 95% confidence interval), and the embedding-vs-pixel comparison persists under augmented re-evaluation. (2) Calibrating ground-truth thresholds using annotations from only three board-certified dermatologists may not fully capture the spectrum of clinical judgment and could introduce biases into our final labels. We provide a threshold sensitivity analysis in Appendix G to quantify the robustness of our choice. (3) As the benchmark builds on a medical imaging data collection, it is challenging and less likely to be saturated quickly. However, data-cleaning strategies performing well on CleanPatrick need to address traits characteristic of medical imaging, such as the importance of fine-grained details or long-tailed labels. These traits are likely more challenging to obtain and require sophisticated methodologies.

**Broader impact and intended use.** CleanPatrick enables rigorous benchmarking of data-cleaning algorithms and supports human-in-the-loop audit workflows. The label-error annotations identify *visually implausible annotations* under conservative human judgment, making them well-suited for: benchmarking data-cleaning algorithms, studying annotation quality and inter-rater reliability, and prioritizing samples for efficient expert review. As with any benchmark derived from crowd annotations, users should interpret results in the context of the task's protocol; our annotations reflect visual incompatibility assessments optimized for precision, complementing other evaluation approaches that may target different aspects of label quality.

## Appendix B. Integration of novel methods

The ReadMe of the GitHub repository[3] details how novel methods can be integrated into the existing benchmark. The integration requires following a minimal adaptation of the existing data cleaning interface, which features methods for detecting the respective quality issues. We encourage people to create pull requests with novel methods to ensure a fair and transparent benchmark.

---

3. github.com/Digital-Dermatology/CleanPatrick, accessed on 26th of September 2025.

## Appendix C. Evaluated approaches

We evaluated different approaches to detect each of the three data quality issue categories, *i.e.*, off-topic samples, near duplicates, and label errors. Some of these methods require encoding images in a low-dimensional latent space. For this projection, we used a vision transformer tiny pre-trained with supervision on ImageNet throughout the paper. In this section, we briefly summarize each evaluated approach, referring, however, to the original paper for more details. All hyperparameters for the evaluated approaches were kept to the default value.

### C.1 Approaches for off-topic samples

**Isolation Forest (IForest)** isolates observations by randomly selecting a feature and splitting the value between the minimum and maximum of the selected feature. The number of splits required to isolate a sample corresponds to the path length from the root node to the leaf node in a tree (Liu et al., 2008). This path length, averaged over a forest of random trees, is a measure of normality, where noticeably shorter paths are produced for anomalies.

**Histogram-based outlier detection (HBOS)** is an efficient unsupervised method that creates a histogram of the feature vector for each dimension and then calculates a score based on how likely a particular data point is to fall within the histogram bins for each dimension (Goldstein and Dengel, 2012). The higher the score, the more likely the data point is an outlier, *i.e.*, a feature vector coming from an anomaly will occupy unlikely bins in one or several of its dimensions and thus produce a higher anomaly score.

**Empirical Cumulative Distribution Functions (ECOD)** is a parameter-free unsupervised outlier detection algorithm (Li et al., 2022). It estimates an empirical cumulative distribution function (ECDF). To generate an outlier score for an observation, it computes the tail probability for each variable using the univariate ECDFs and multiplies them.

**AutoEncoder** is a reconstruction-based unsupervised method (Aggarwal et al., 2015). It learns to compress and then decompress data, and the reconstruction error for a given sample serves as its anomaly score. Samples that the model struggles to reconstruct accurately are considered anomalous.

**Variational AutoEncoder (VAE)** is a generative extension of the AutoEncoder (Kingma and Welling, 2014). Similar to a standard AutoEncoder, it uses reconstruction error as an anomaly score, but its probabilistic latent space can offer a more robust representation for identifying outliers.

**DeepSVDD (Deep Support Vector Data Description)** is a deep one-class classification method that learns a neural network transformation to map most normal data points into a compact hypersphere in the output space (Ruff et al., 2018). The distance from a sample's representation to the center of this hypersphere is used as its anomaly score.

**DIF (Deep Isolation Forest)** is a deep learning-based extension of the classic Isolation Forest algorithm (Xu et al., 2023). It leverages representation learning to enhance the isolation process, making it more effective for complex, high-dimensional data like images.

### C.2 Approaches for near duplicates

**Perceptual Hash (pHashing)** is a type of locality-sensitive hash, which is similar if features of the sample are similar (Venkatesan et al., 2000). It relies on the discrete cosine transform (DCT) for dimensionality reduction and produces hash bits depending on whether each DCT value is above or below the average value. In this paper, we use pHash with a hash size of 8.

**Structural Similarity Index Measure (SSIM)** is a type of similarity measure to compare two images with each other based on three features, namely luminance, contrast, and structure (Wang et al., 2004). Instead of applying SSIM globally, *i.e.*, all over the image at once, one usually applies the metrics regionally, *i.e.*, in small sections of the image, and takes the mean over all. This variant of SSIM is often called "Mean Structural Similarity Index". In this paper, we apply SSIM locally to 8x8 windows but still refer to the method as SSIM for simplicity.

### C.3 Approaches for label errors

**Confident Learning (CLearning)** is a data-centric approach that focuses on label quality by characterizing and identifying label errors in datasets based on the principles of pruning noisy data, counting with probabilistic thresholds to estimate noise, and ranking examples to train with confidence (Northcutt et al., 2021a). It builds upon the assumption of a class-conditional noise process to directly estimate the joint distribution between noisy (given) and uncorrupted (unknown) labels, resulting in a generalized learning process that is provably consistent and experimentally performant. In this study, we use AdaBoost (Freund et al., 1999) as a classifier on top of pre-trained representations to estimate probabilities. We did not observe any significant performance difference when using different classifiers similarly to Northcutt et al. (2021a).

**NoiseRank (Noise)** is a method for unsupervised label noise detection using Markov Random Fields (Sharma et al., 2020). It constructs a dependence model to estimate the posterior probability of an instance being incorrectly labeled, given the dataset, and ranks instances based on this probability.

**FINE (Finding Clean Samples for Learning with Noisy Labels)** is a method designed to identify correctly labeled ("fine") samples within a noisy dataset (Kim et al., 2021). It aims to distinguish the clean subset from the noisy one, enabling more robust model training.

**BHN (Delving into Noisy Label Detection with Clean Data)** is a noisy label detection method that leverages a small set of clean, validated data to improve the identification of mislabeled examples in the larger, noisy dataset (Yu et al., 2023).

### C.4 Approaches for multiple issue types

**SelfClean** leverages context-aware self-supervised embeddings learned on the contaminated dataset and employs simple distance-based indicators in that latent space, *i.e.*, clustering for off-topic detection, nearest-neighbor distances for near-duplicates, and class-wise distance comparisons for label errors, to rank and score samples for inspection (Gröger et al., 2024). The methodology is intended to be used with a human in the loop, where top-ranked issues are validated. However, it can also be used fully automatically by thresholding based on estimated contamination.

## Appendix D. Details to Annotation Platform

Figure 4 shows a screenshot of the annotation platform of Centaur Labs, specifically of the DiagnosUs app[4], used for obtaining annotations. Additionally, it also shows the instructions given to the medical crowd workers and expert annotators.

DiagnosUs is a free app where annotators voluntarily opt in to contests, where medical image annotations are completed by labelers competing in these challenges. Labelers' submissions are scored based on their accuracy against a set of gold standard cases. Labelers who achieve high accuracy and are placed on the leaderboard are compensated with monetary prizes. Prize amounts vary depending on the contest structure, ranging from approximately \$0.50 to \$20 per prize.

---

4. `https://www.diagnosus.com/`, accessed on 26th of September 2025.

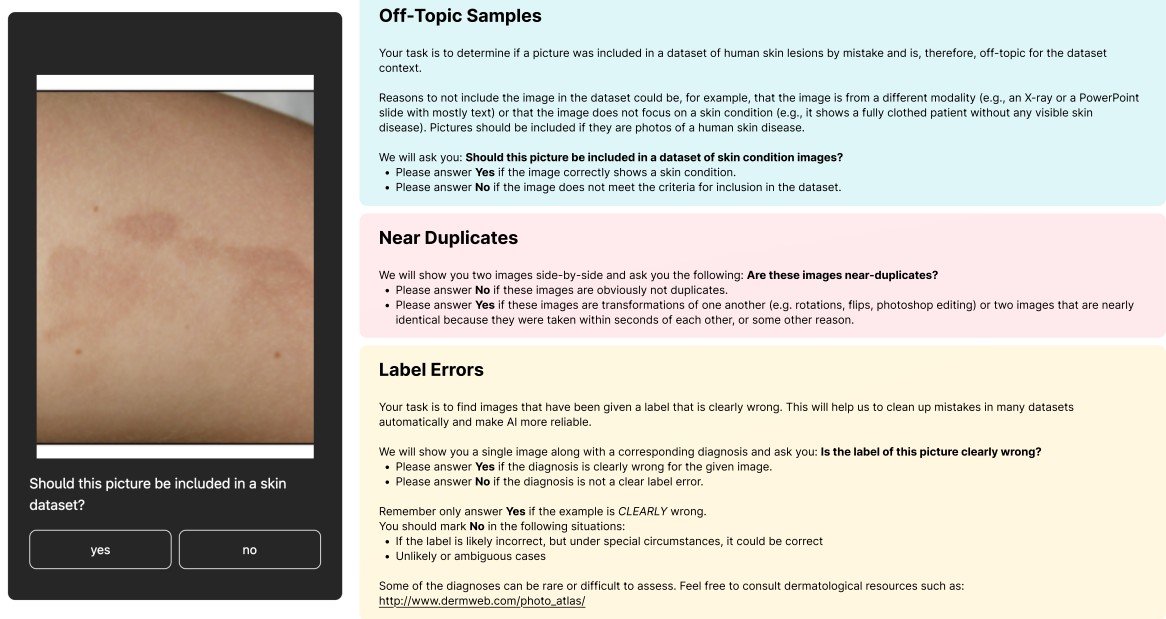

Figure 4: Left shows a screenshot of the labeling interface shown to the medical crowd workers. Right shows the instructions given to the annotators for the respective labeling tasks for the data quality issues. Along with each set of instructions, the annotators were given some example images of both the positive and negative responses.

## Appendix E. Detailed analysis of data quality issues

**Annotation counts.**    Figure 5 illustrates the distribution of annotation counts per sample across the three issue types: off-topic samples, near duplicates, and label errors. On average, each image received 10 medical crowd worker votes, with extremes ranging from a single annotation to as many as 225. The vast majority of samples fall between 5 and 20 annotations.

Notably, *only one sample* ended up with a single annotation. This occurred because a medical crowd worker mistakenly flagged the image early in the process, causing it to be excluded from subsequent annotation rounds. To ensure no gap in quality, the authors manually reviewed this outlier in full and confirmed its correct classification in the final benchmark.

**Near-duplicate components.**    Beyond per-sample vote counts, we also examined how near-duplicate samples group into connected components under our fast-duplicate detection procedure. We discovered 2,389 separate components of size $\geq 2$. Their size distribution is shown in Table 2. This distribution shows that small components (pairs and triplets) dominate the duplicate structure, while only a handful of larger clusters (size $\geq 10$) exist.

## Appendix F. Fast near duplicates

**Near duplicates.**    Let $\mathcal{D} = \{1, 2, \ldots, |\mathcal{D}|\}$ be a dataset with samples labeled by consecutive integers. A sample pair $\{i, j\}$ corresponds to an edge in a graph with vertices $\mathcal{D}$. Verifying all near-duplicate pairs in $\mathcal{D}$ is equivalent to annotating each edge in the complete graph, and yields the subgraph $\mathcal{D}_\sim$ induced by the binary near-duplicate relation $\sim$.

Table 2: Counts of near-duplicate components by size

| Component Size | Number of Components |
|:---:|:---:|
| 2 | 1997 |
| 3 | 169 |
| 4 | 151 |
| 5 | 19 |
| 6 | 26 |
| 7 | 8 |
| 8 | 9 |
| 10 | 4 |
| 11 | 2 |
| 12 | 2 |
| 25 | 1 |
| 30 | 1 |

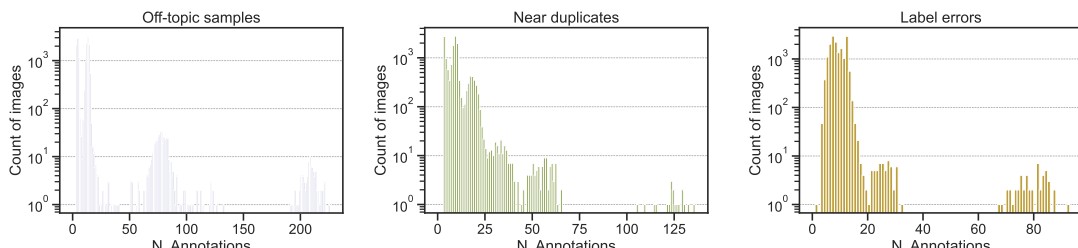

Figure 5: Histograms showing the number of annotations from medical crowd workers per image sample for each data quality issue.

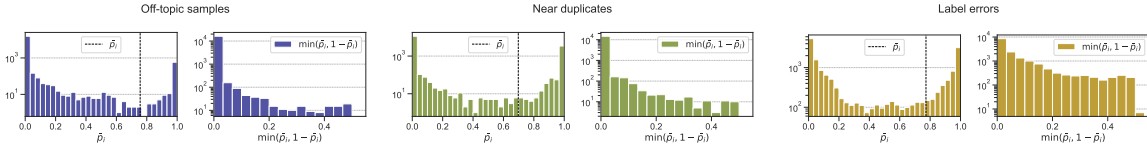

Figure 6: Distribution of $\bar{p}_i$ (left panels) and per-sample annotation uncertainty $\min\big(\bar{p}_i,\, 1 - \bar{p}_i\big)$ (right panels) for the three issue types: off-topic samples, near duplicates, and label errors. Here, $\bar{p}_i$ is the estimated probability that sample $i$ belongs to the positive class under the GLAD model. Vertical dashed lines in the left panels indicate the expert-calibrated thresholds $t_{\mathrm{OT}} = 0.76$, $t_{\mathrm{ND}} = 0.72$, and $t_{\mathrm{LE}} = 0.77$ used to produce the final labels.

**Automatic cleaning.** Consider a function $d : \mathcal{D} \times \mathcal{D} \to \mathbb{R}$ that associates a unique weight $d(i, j)$ to each edge, $d(i, j) \neq d(k, l)$ for any $i, j, k, l \in \mathcal{D}$. To keep the language and notation intuitive, we assume $d$ is a distance as is the case in this work, but one can still obtain a similar procedure *mutatis mutandis* if this is not the case. Near-duplicate detection could be easily performed exactly if the following property were to hold.

**Assumption 1 (Automatic Cleaning)** *All near-duplicate pairs have a distance which is less than any non-near-duplicate pair, i.e.*

$$\forall i, j, k, l \in \mathcal{D} \mid i \sim j \wedge k \nsim l, \quad d(i,j) < d(k,l). \tag{1}$$

In this case, there is a threshold $d_*$ such that all near-duplicate pairs have a smaller distance, and any pair with larger distance is not a near duplicate. In practice, such a perfect ranking is very difficult to find, and one has to resort to hybrid methods which require human verification. This generates the burden of annotating all sample pairs, which grow quadratically with the dataset size.

**Fast cleaning.**  To determine which samples are potentially related to each other, it is sufficient to partition the samples according to which subgraph they belong to, without necessarily knowing every pairwise relation. For this task, the poor scaling of manual verification can be significantly alleviated with the help of a function that satisfies the following, weaker condition.

**Assumption 2 (Fast cleaning)** *Near duplicates of a sample are closer to it than other samples, i.e.*

$$\forall i, j, k \in \mathcal{D} \mid i \sim j \wedge i \nsim k, \quad d(i,j) < d(i,k). \tag{2}$$

As a heuristic side note, we observe that this assumption is substantially more *local* than assumption 1, as it only requires the distance to correctly sort near duplicates in the neighborhood of the specific sample $i$.

To exploit the fast cleaning assumption, one may proceed analogously to Borůvka's algorithm for minimal spanning trees (Borůvka, 1926). For every sample $i \in \mathcal{D}$, find its nearest neighbor $n(i) \in \mathcal{D} \setminus \{i\}$ to build the set of neighbor pairs $\mathcal{N} = \{i, n(i)\}_{i \in \mathcal{D}}$. Checking all of them takes $|\mathcal{N}|$ annotations and gives the set of near duplicates $\mathcal{P}$. By virtue of assumption 2, all samples which do not appear in $\mathcal{P}$ have no near duplicates. The pairs in $\mathcal{P}$, instead, are edges that belong to the subgraph $\mathcal{D}_\sim$. The connected components of $\mathcal{P}$ partition its vertices in a set of clusters $\mathcal{D}_1$. Because of assumption 2, two such subsets belong to the same connected component of $\mathcal{D}_\sim$ if and only if their two elements with the smallest distance are near duplicates. Therefore, it is now sufficient to annotate the nearest-neighbor pairs $\mathcal{N}_1$ within $\mathcal{D}_1$ and iterate the procedure. This is guaranteed to exactly identify the connected components of $\mathcal{D}_\sim$ once no more near duplicates are found.

**Cleaning complexity.**  When $d$ is symmetric and there are no ties, the number of connected components in the subgraph of nearest neighbors $\mathcal{N}$ is exactly equal to $|\mathcal{D}| - |\mathcal{N}|$, *i.e.*, the number of duplicated nearest-neighbor edges. Indeed, each sample appears in at least one edge, so it belongs to a component which connects it to its nearest neighbor, then to the nearest neighbor thereof, and so on. However, there can be no cycles in the nearest neighbor subgraph $\mathcal{N}$, else the first sample would have connected to the last instead of the second. Any such tree therefore terminates with two samples which are reciprocally the closest to each other. The number of undirected edges that appear twice in the list $\{i, n(i)\}_{i \in \mathcal{D}}$ is therefore the number of connected components in $\mathcal{N}$, and can be expressed as $|\mathcal{D}| - |\mathcal{N}|$.

After the $i$-th iteration, one can scan the verified near-duplicate clusters $\mathcal{D}_i$ obtained in the last step (which have size larger than $2^i$), and find the set $\mathcal{N}_i$ of the closest sample pairs that belong to different clusters. The number of nearest neighbors pairs to verify is $|\mathcal{N}_i|$ and leaves $|\mathcal{D}_{i+1}| \leq |\mathcal{D}_i| - |\mathcal{N}_i|$ new clusters that require another iteration. This terminates when the $k$-th iteration generates no new subsets, $|\mathcal{D}_k| = 0$. Since the size of the new subgraphs $\mathcal{D}_i$ is at least double at each iteration, one has $k \leq \lfloor \log_2 K \rfloor + 1$ where $K$ is the size of the largest subgraph and clearly $K \leq |\mathcal{D}|$. Manipulating inequalities to have the $|\mathcal{N}_i|$ terms on the left side, the total number of annotations clearly satisfies

$$|\mathcal{N}| + |\mathcal{N}_1| + |\mathcal{N}_2| + \cdots + |\mathcal{N}_{k-1}| \leq (|\mathcal{D}| - |\mathcal{D}_1|) + (|\mathcal{D}_1| - |\mathcal{D}_2|) + \cdots + |\mathcal{D}_{k-1}| = |\mathcal{D}|. \tag{3}$$

We thus have the following guarantee:

**Lemma 1** *Finding all near duplicate clusters under assumption 2 requires annotating at most $|\mathcal{D}|$ sample pairs in at most $\lfloor \log_2 K \rfloor + 1$ iterations.*

**Comment on transitivity.** One may be tempted to think that near duplicates correspond to an equivalence relation as follows.

**Assumption 3 (Near-duplicate equivalence)** *The near-duplicate relation $\sim$ satisfies*

1. $i \sim i$ *(reflexive)*
2. $i \sim j \Rightarrow j \sim i$ *(symmetric)*
3. $i \sim j \wedge j \sim k \Rightarrow i \sim k$ *(transitive)*

*for any $i, j, k \in \mathcal{D}$.*

However, this is clearly not true in practice. A counterexample are the frames of a video that was captured without interruptions but features two very different situations at the beginning and at the end. While each two consecutive frames are near duplicates, it is a question if the first and the last frames taken alone should be considered near duplicates. For this reason, it is in general better to always consider merging clusters based on the two most similar samples, *i.e.*, using single linkage.

## Appendix G. Threshold Sensitivity Analysis

To assess the sensitivity of our benchmark labels to the expert-calibrated threshold choice, we examine how the estimated prevalence of each issue type varies as the binarization threshold changes around the calibrated values ($t_{\mathrm{OT}} = 0.76$, $t_{\mathrm{ND}} = 0.72$, $t_{\mathrm{LE}} = 0.77$). For each issue type, we vary the threshold $t$ by $\pm 0.15$ around the calibrated value and report the resulting number of positive samples. This analysis demonstrates that the sensitivity of labels to the calibrated thresholds is negligible for both near duplicates and label errors, and weak for label errors.

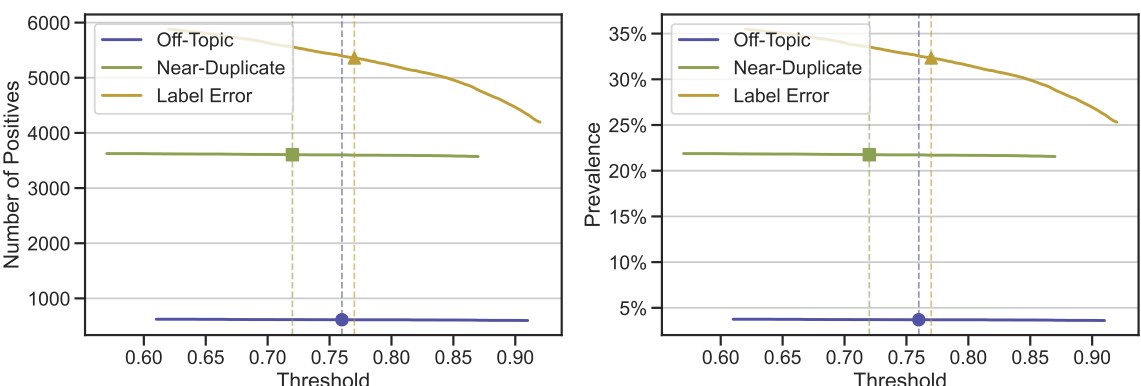

Figure 7: Effect of threshold variation on estimated prevalence (number of positive samples) for each data quality issue type. Vertical dashed lines indicate the expert-calibrated thresholds used in the benchmark ($t_{\mathrm{OT}} = 0.76$, $t_{\mathrm{ND}} = 0.72$, $t_{\mathrm{LE}} = 0.77$).

**Results.**

**Bootstrap over annotators.** We also quantify the uncertainty on the threshold itself by bootstrapping the panel of three medical experts. For each task, we resample three experts with replacement $B = 1{,}000$ times, recompute the majority labels on the 400 quantile-stratified samples, and re-apply the bin-mean calibration rule. The resulting bootstrap distribution of the threshold is

shown in Figure 8. The threshold is invariant across all 1,000 iterations for off-topic samples and near-duplicates, because the resampled panel always agrees on the small set of positive samples that drive the rule. For label errors the threshold has a wider distribution as expected because of the lower agreement, with the bin-mean value 0.7727 selected in roughly half of iterations and lower bins in the remainder. Across the 95% percentile interval [0.5053, 0.7727] the number of label-error positives ranges from about 5,300 to 6,500. The relative ranking of label-error detectors is unchanged across this range, as shown in Appendix I.

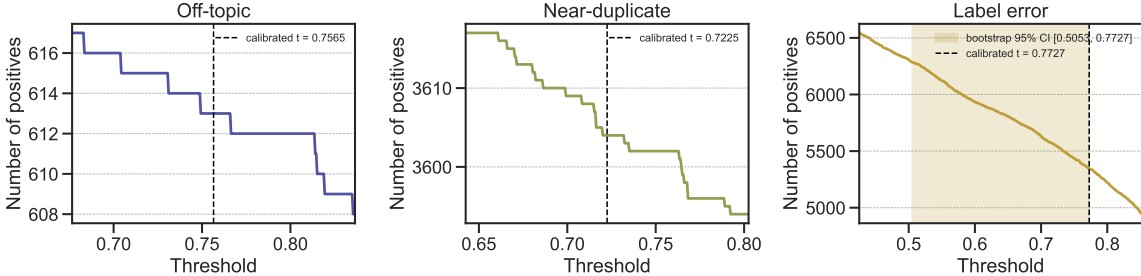

Figure 8: Bootstrap over annotators. Number of positives as a function of the binarization threshold for each task. The dashed line marks the algorithm-calibrated threshold under the original 3-expert majority. The shaded band on the label-error panel is the 95% percentile interval of the threshold under bootstrap resampling of the three experts with replacement ($B = 1,000$ iterations). The bootstrap collapses to a single threshold for off-topic samples and near-duplicates, and produces a wider interval for label errors consistent with the lower expert agreement on this task.

## Appendix H. Duplicate-retrieval bias audit

The near-duplicate ground truth in CleanPatrick is constructed via embedding-based candidate retrieval (ImageNet-pretrained DINO) followed by human verification. To bound the bias this introduces against methods based on different representations, we consider other retrieval criteria.

**Sampling.** For three representations that span a competing hypothesis space, *i.e.*, perceptual hashing (pHash), structural similarity (SSIM), and an ImageNet-supervised ViT-T, we ranked all $\binom{16,574}{2}$ image pairs by similarity, removed pairs that were already retrieved by ImageNet-DINO, and randomly sampled 150 of each method's top 5,000 pairs. The pools were sampled jointly without replacement, yielding 450 unique pairs from regions of high similarity not surfaced by the original DINO-based candidate retrieval.

**Annotation.** Three domain experts independently labeled each of the 450 pairs as *yes* (near-duplicate), *no*, or *uncertain*, following the same protocol used for the original near-duplicate annotation campaign. Inter-annotator agreement was strong, with a mean pairwise Cohen's $\kappa$ of 0.89 computed across the three response categories. This is a strict convention, because *uncertain* counts as a distinct category for agreement. Uncertain responses are rare in practice, with only 9 of the 450 pairs that received any at all. We treat *uncertain* pairs as not a near-duplicate when forming the audit label. Under the alternative conventions (dropping uncertain pairs or counting them toward duplicates) the pHash and SSIM counts remain zero, and the ImageNet ViT-T count varies between 26 and 32 out of 150.

**Audit results.** Table 3 reports the number of confirmed duplicates per source representation. None were found among the 300 audited pairs sampled from pHash and SSIM, so the Wilson 95%

Table 3: Candidate-retrieval bias audit. For each competing representation, we randomly sampled 150 of its top-5,000 most-similar non-candidate pairs and obtained three expert annotations per pair (mean pairwise Cohen's $\kappa = 0.89$). *Strict* requires unanimous agreement. *Majority* requires more than half. Wilson 95% CIs in brackets.

| Source representation | Audited | Strict dup. | Majority dup. |
|---|---|---|---|
| pHash | 150 | 0 (0.0 [0.0, 2.5]%) | 0 (0.0 [0.0, 2.5]%) |
| SSIM | 150 | 0 (0.0 [0.0, 2.5]%) | 0 (0.0 [0.0, 2.5]%) |
| ImageNet ViT-T (sup.) | 150 | 24 (16.0 [11.0, 22.7]%) | 26 (17.3 [12.1, 24.2]%) |

Table 4: Near-duplicate benchmark on three evaluation pools: the original candidate-pair pool from the paper, the same pool augmented with the 17 audit-confirmed duplicates not already present via clique transitivity (*+positives only*), and the same pool augmented with all 450 audited pairs (*+full*). Values are mean (%) with bootstrap 95% CIs from 1,000 samples.

| Pool | Method | AUROC | AP | P@100 | P@1000 |
|---|---|---|---|---|---|
| Original (paper) | pHash | 49.1 [48.3, 50.1] | 30.7 [29.8, 31.7] | 32.0 | 30.3 |
| | SSIM | 49.1 [48.1, 50.0] | 30.5 [29.5, 31.5] | 31.0 | 27.6 |
| | SelfClean | 91.7 [91.2, 92.1] | 87.9 [87.2, 88.5] | 100.0 | 100.0 |
| + positives only | pHash | 49.3 [48.3, 50.2] | 31.1 [30.0, 32.1] | 38.0 | 30.9 |
| | SSIM | 49.2 [48.2, 50.2] | 30.8 [29.9, 31.8] | 37.0 | 28.7 |
| | SelfClean | 91.5 [91.0, 92.0] | 87.7 [87.1, 88.4] | 100.0 | 100.0 |
| + full (450 pairs) | pHash | 48.2 [47.3, 49.1] | 28.7 [27.9, 29.5] | 0.0 | 20.9 |
| | SSIM | 47.9 [47.0, 48.9] | 28.3 [27.5, 29.1] | 1.0 | 20.1 |
| | SelfClean | 91.8 [91.3, 92.3] | 87.7 [86.9, 88.4] | 100.0 | 100.0 |

confidence interval gives a maximum prevalence of 2.5% in each. The supervised ViT-T audit recovered 26 majority-confirmed duplicates among 150 pairs (17.3%, Wilson 95% CI [12.1%, 24.2%], of which 24 had unanimous agreement). The comparison between embeddings and pixel methods is therefore robust to retrieval bias.

**Benchmark stability.** We re-evaluate all near-duplicate detectors on two variations of the original evaluation pool. The (*+full*) variant adds all 450 audited pairs with their majority labels. The (*+positives only*) variant adds only the 17 confirmed duplicates that were not already identified via clique transitivity. Table 4 summarizes the results. SelfClean's AUROC remains stable while its lead over pixel-based methods widens slightly, as pHash and SSIM each lose 1–2 AUROC points because of high-scoring pairs that are not duplicates.

**Recall on duplicates outside the candidate pool.** Restricted to the 17 audit-confirmed duplicates that fell outside the original candidate pool, SelfClean recovers 76.5% at a review budget of $k = 5,000$, compared to 17.6% and 35.3% for pHash and SSIM, respectively (Table 5). Self-supervised representations therefore retain an advantage on duplicates that the candidate-retrieval representation itself failed to surface.

Table 5: Recall at top-$k$ on the 17 confirmed-duplicate non-candidate pairs (*i.e.*, duplicates outside the original DINO-based candidate pool, identified by the audit).

| Method | R@100 | R@500 | R@1000 | R@2000 | R@5000 |
|---|---|---|---|---|---|
| pHash | 0.0% | 5.9% | 5.9% | 5.9% | 17.6% |
| SSIM | 0.0% | 0.0% | 0.0% | 5.9% | 35.3% |
| SelfClean | 0.0% | 0.0% | 5.9% | 11.8% | 76.5% |

## Appendix I. Sensitivity to expert-level disagreement for label errors

The label-error inter-annotator agreement among experts ($\alpha = 0.42 \pm 0.06$) sits at the boundary between fair and moderate agreement on the conventional Landis–Koch scale (Landis and Koch, 1977), below the $\alpha \geq 0.667$ threshold typically used as a reliability cutoff. Such clinical disagreement is known to propagate into model evaluation (Lionetti et al., 2025). To quantify its potential impact, we conduct two complementary sensitivity analyses on the 400 quantile-stratified samples annotated by the three experts. These are a deterministic leave-one-expert-out (LOO) check and a hierarchical bootstrap over experts and samples. Note that the labels evaluated in this appendix are the 400-sample expert labels, not the benchmark labels released with CleanPatrick. The released labels are derived from the IRT model and the calibration threshold, which is robust to expert perturbations as shown in Appendix G. Furthermore, the benchmark itself is not even sensitive to the threshold, as the main metrics AUROC and AP only depend on the ranking. The 400-sample expert subset is the only place where expert-level disagreement directly changes the ground truth, and the analyses below study the impact on that subset.

**Leave-one-expert-out.** For each fold (the original 3-expert majority and the three LOO majorities-of-2, with ties broken to non-error), we re-evaluate each label-error detector against the fold-specific 400-sample labels, using the per-sample scores from the original benchmark run. SelfClean attains the highest AUROC under all four labelings (baseline 55.8, and 56.0, 59.6, 57.0 across the three LOO folds), and the relative ordering of detectors is preserved: SelfClean > ConfidentLearning ≈ BHN > NoiseRank in every fold. Cohen's $\kappa$ between the LOO labels and the 3-expert baseline is 0.87–0.95, which means that single-expert removal perturbs only a small fraction of the 400 labels as expected.

**Hierarchical bootstrap.** We complement the LOO check with a hierarchical bootstrap of $B = 1,000$ iterations. Each iteration resamples three experts with replacement to form a panel, resamples 400 samples with replacement to form an evaluation subset, aggregates the resampled experts' votes per sample to obtain the iteration's labels, and recomputes AUROC and AP. The intervals in Table 7 therefore combine expert-panel and sample-level uncertainty. The 95% intervals overlap due to the small evaluation pool of 400 expert-annotated samples and the limited expert panel of three raters. Bootstrap means preserve the ranking observed in the main paper (SelfClean > ConfidentLearning ≈ BHN > NoiseRank).

## Appendix J. Annotation-count sensitivity

The IRT model used to aggregate crowd annotations yields posterior probabilities $\bar{p}_i$ whose uncertainty depends on the number and value of annotations per sample (or per pair, for near-duplicates). While the average is approximately 10 votes per item, the distribution is right-skewed and a fraction of items receive few annotations. This appendix evaluates whether the benchmark is robust to excluding low-vote items.

Table 6: Leave-one-expert-out sensitivity for the label-error sub-task. Each fold uses the majority of the remaining two experts (tie → non-error). Cohen's $\kappa$ measures the agreement of the 400 fold-specific labels with the original 3-expert majority.

| Method | Held-out expert | $n_+$ | $\kappa$ | AUROC [%] | AP [%] |
|---|---|---|---|---|---|
| SelfClean | — (3-expert) | 33 | — | 55.8 | 17.0 |
| SelfClean | expert 1 | 30 | 0.95 | 56.0 | 16.8 |
| SelfClean | expert 2 | 26 | 0.87 | 59.6 | 16.4 |
| SelfClean | expert 3 | 27 | 0.89 | 57.0 | 15.7 |
| ConfidentLearning | — (3-expert) | 33 | — | 55.2 | 9.9 |
| ConfidentLearning | expert 1 | 30 | 0.95 | 52.4 | 8.6 |
| ConfidentLearning | expert 2 | 26 | 0.87 | 54.5 | 8.1 |
| ConfidentLearning | expert 3 | 27 | 0.89 | 55.6 | 8.6 |
| BHN | — (3-expert) | 33 | — | 51.0 | 10.0 |
| BHN | expert 1 | 30 | 0.95 | 49.6 | 9.4 |
| BHN | expert 2 | 26 | 0.87 | 52.3 | 9.2 |
| BHN | expert 3 | 27 | 0.89 | 52.5 | 9.4 |
| NoiseRank | — (3-expert) | 33 | — | 43.7 | 7.2 |
| NoiseRank | expert 1 | 30 | 0.95 | 47.2 | 7.0 |
| NoiseRank | expert 2 | 26 | 0.87 | 46.4 | 6.2 |
| NoiseRank | expert 3 | 27 | 0.89 | 44.8 | 6.3 |

Table 7: Hierarchical bootstrap over experts and samples for the label-error sub-task. For each of $B = 1,000$ iterations, we resample 3 experts with replacement (panel) and 400 samples with replacement (subset), and recompute AUROC and AP using the per-iteration majority labels and the original benchmark scores. Reported are bootstrap means with 95% percentile intervals that combine expert-panel and sample-level uncertainty.

| Method | AUROC [%] | AP [%] |
|---|---|---|
| SelfClean | 56.2 [43.5, 67.2] | 22.1 [9.0, 40.5] |
| ConfidentLearning | 51.8 [42.0, 62.4] | 15.4 [6.8, 33.2] |
| BHN | 51.6 [41.3, 61.0] | 16.8 [7.2, 32.1] |
| NoiseRank | 44.8 [34.3, 53.3] | 12.7 [5.3, 28.0] |

**Protocol.** For each task and each minimum-vote cutoff $k \in \{1, 3, 5, 7, 10, 15\}$, we restrict the evaluation to items with at least $k$ annotators and recompute AUROC and AP on this subset using the per-item detector scores from the original benchmark run. Coverage at each cutoff is summarised in Table 8. The AUROC trajectory per task and method is shown in Figure 9 and Table 9.

**Ranking stability.** Across all three tasks, the relative ranking of methods is preserved for any cutoff $k \leq 10$ (Figure 9, Table 9). For off-topic and near-duplicate detection, absolute AUROC values remain within $\pm 1$ percentage point of the unrestricted pool ($k = 1$) at every cutoff: SelfClean stays in the 97.5–98.3 AUROC range on near-duplicates with a gap to pixel-based methods that exceeds 45 AUROC points, and IForest leads SelfClean by approximately 10 AUROC points on

Table 8: Fraction of items kept above each minimum-vote cutoff $k$ for the three issue types. Off-topic and label-error counts are per sample. Near-duplicate counts are per directly-annotated pair.

| Task | $k \geq 1$ | $k \geq 3$ | $k \geq 5$ | $k \geq 7$ | $k \geq 10$ | $k \geq 15$ |
|---|---|---|---|---|---|---|
| Off-topic | 100.0% | 100.0% | 69.3% | 61.9% | 61.3% | 7.2% |
| Label error | 100.0% | 100.0% | 97.5% | 78.6% | 39.2% | 1.1% |
| Near-duplicate (pairs) | 100.0% | 100.0% | 75.6% | 69.6% | 34.7% | 17.2% |

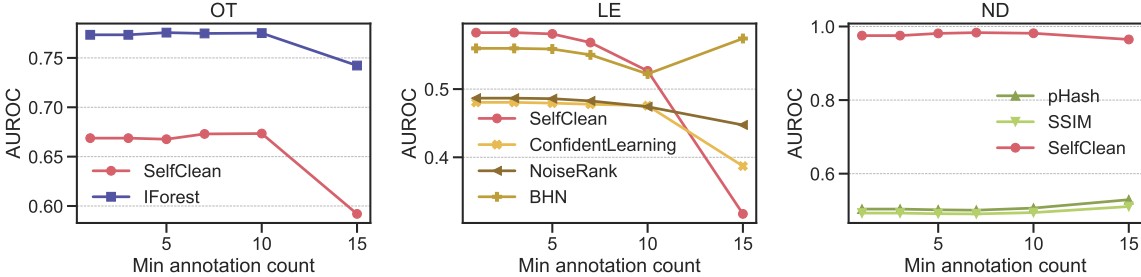

Figure 9: AUROC versus minimum-vote cutoff for the top methods of each task. Markers denote the cutoffs in Table 8. The relative ranking is preserved for any cutoff $k \leq 10$ across all three tasks. The drop on the label-error panel at $k = 15$ is a small-sample artefact discussed in the text.

off-topic detection. For label errors, ConfidentLearning and NoiseRank stay within $\pm 1$ point of the unrestricted pool, while the top two methods (SelfClean and BHN) drift down by 3–6 points at $k = 10$. This drift is driven by the composition of the high-vote tail, which is dominated by samples on which the annotation platform allocated more annotators before reaching consensus and which are therefore the most ambiguous. SelfClean still leads BHN at every cutoff, and the SelfClean–BHN pair still leads ConfidentLearning and NoiseRank by 5–10 AUROC points throughout $k \leq 10$.

**The label-error tail.** At $k = 15$, only 1.1% of label-error samples are retained. This tail comprises items on which the annotation platform allocated the most annotators before reaching consensus. By construction these are the most disagreement-prone borderline cases, and the resulting subset is too small to support stable ranking estimates.

Table 9: AUROC (%) versus minimum-vote cutoff $k$ for the top methods of each task. The relative ranking is preserved for any cutoff $k \leq 10$ across all three tasks. The LE flip at $k = 15$ is on a tail of $\sim 180$ samples (1.1% of the total) with the highest annotator count, which are by construction cases with high disagreement.

| Task | Method | $k \geq 1$ | $k \geq 3$ | $k \geq 5$ | $k \geq 7$ | $k \geq 10$ | $k \geq 15$ |
|------|--------|------|------|------|------|------|------|
| OT | SelfClean | 66.9 | 66.9 | 66.8 | 67.3 | 67.3 | 59.2 |
|    | IForest | 77.3 | 77.3 | 77.6 | 77.5 | 77.5 | 74.2 |
| LE | SelfClean | 58.3 | 58.3 | 58.1 | 56.8 | 52.7 | 31.7 |
|    | BHN | 56.0 | 56.0 | 55.9 | 55.0 | 52.2 | 57.4 |
|    | ConfidentLearning | 48.1 | 48.1 | 47.9 | 47.8 | 47.6 | 38.7 |
|    | NoiseRank | 48.7 | 48.7 | 48.6 | 48.3 | 47.4 | 44.7 |
| ND | pHash | 50.4 | 50.4 | 50.2 | 50.1 | 50.6 | 52.9 |
|    | SSIM | 49.3 | 49.3 | 49.1 | 49.1 | 49.4 | 51.1 |
|    | SelfClean | 97.5 | 97.5 | 98.1 | 98.3 | 98.2 | 96.5 |

## Appendix K. Detailed plots and results

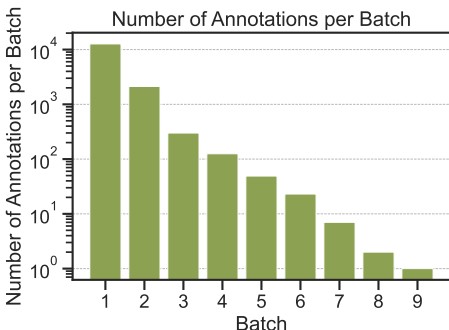

Figure 10: Number of duplicates per batch of annotation. For each batch, we select the closest pairs which has been positively identified as near duplicates in the batch before and start by taking the closest pair for every sample in the dataset.

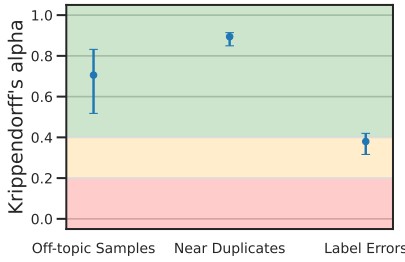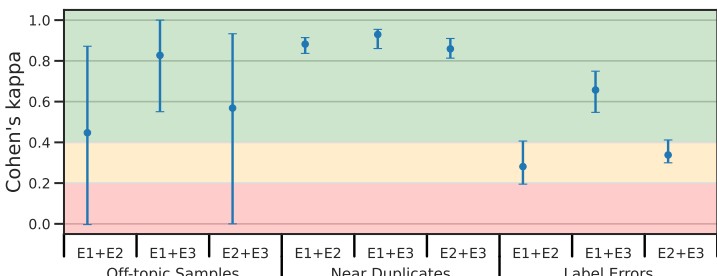

Figure 11: Inter-annotator agreement as Krippendorff's alpha among all expert annotators (left) and Cohen's kappa for all expert annotator pairs (right). Markers identify the three data-quality issue types, error bars are 95% confidence intervals obtained by bootstrapping annotated samples, and the background color indicates the degree of agreement (Regier et al., 2013).

Table 10: Detailed performance of evaluated approaches on CleanPatrick. Consult appendix C for details on competing approaches. Values are reported as percentages (%) and represent the mean with 95% confidence intervals obtained via 2,000 bootstrap samples.

| | Method | $p^+$ | P@100 | P@500 | P@1000 | R@100 | R@500 | R@1000 | AUROC | AP |
|---|---|---|---|---|---|---|---|---|---|---|
| Off-Topic Samples | kNN | 3.7 | 25.0 [17.6–34.3] | 15.0 [12.1–18.4] | 9.9 [8.2–11.9] | 4.1 [2.7–5.7] | 12.2 [9.8–14.6] | 16.2 [13.5–19.2] | 63.6 [61.3–65.9] | 8.5 [7.0–10.5] |
| | IForest | 3.7 | 39.0 [30.0–48.8] | 22.6 [19.2–26.5] | 18.1 [15.8–20.6] | 6.4 [4.8–8.0] | 18.4 [15.8–21.4] | 29.5 [26.3–32.9] | 77.3 [75.4–79.2] | 15.9 [13.5–18.9] |
| | HBOS | 3.7 | 37.0 [28.2–46.8] | 20.2 [16.9–23.9] | 16.7 [14.5–19.1] | 6.0 [4.7–7.9] | 16.5 [13.8–19.1] | 27.2 [23.9–30.4] | 75.5 [73.4–77.5] | 15.2 [12.7–18.1] |
| | ECOD | 3.7 | 38.0 [29.1–47.8] | 21.2 [17.8–25.0] | 17.4 [15.2–19.9] | 6.2 [4.8–8.0] | 17.3 [14.7–20.3] | 28.4 [24.9–31.8] | 75.7 [73.6–77.8] | 16.2 [13.6–19.3] |
| | AutoEncoder | 3.7 | 41.0 [31.9–50.8] | 19.6 [16.4–23.3] | 15.0 [12.9–17.4] | 6.7 [5.0–8.1] | 16.0 [13.2–18.7] | 24.5 [21.3–27.7] | 72.7 [70.6–74.8] | 12.6 [10.5–15.2] |
| | DeepSVDD | 3.7 | 36.0 [27.3–45.8] | 19.2 [16.0–22.9] | 15.0 [12.9–17.4] | 5.9 [4.4–7.6] | 15.7 [13.1–18.4] | 24.5 [21.4–27.7] | 72.8 [70.7–74.8] | 13.0 [10.9–15.6] |
| | VAE | 3.7 | 40.0 [30.9–49.8] | 21.6 [18.2–25.4] | 17.0 [14.8–19.5] | 6.5 [4.9–8.0] | 17.6 [15.0–20.3] | 27.7 [24.5–31.3] | 75.6 [73.5–77.5] | 15.2 [12.9–18.1] |
| | DIF | 3.7 | 34.0 [25.5–43.7] | 21.0 [17.7–24.8] | 16.5 [14.3–18.9] | 5.6 [4.0–7.1] | 17.1 [14.3–19.7] | 26.9 [23.5–30.3] | 73.2 [71.1–75.3] | 12.7 [10.8–15.2] |
| | SelfClean | 3.7 | 52.0 [42.3–61.5] | 21.0 [17.7–24.8] | 13.1 [11.2–15.3] | 8.5 [6.8–10.3] | 17.1 [14.8–20.4] | 21.4 [18.2–24.6] | 66.9 [64.9–68.7] | 14.5 [11.9–17.3] |
| | Method | $p^+$ | P@100 | P@500 | P@1000 | R@100 | R@500 | R@1000 | AUROC | AP |
| Near Duplicates | pHashing | 21.4 | 36.0 [27.3–45.8] | 31.0 [27.1–35.2] | 31.8 [29.0–34.8] | 0.7 [0.5–0.9] | 2.9 [2.5–3.3] | 6.0 [5.5–6.6] | 50.5 [49.6–51.5] | 31.6 [30.6–32.5] |
| | SSIM | 21.4 | 31.0 [22.8–40.6] | 28.0 [24.2–32.1] | 27.6 [24.9–30.5] | 0.6 [0.4–0.8] | 2.7 [2.3–3.0] | 5.2 [4.7–5.8] | 49.1 [48.1–50.0] | 30.5 [29.5–31.4] |
| | SelfClean | 21.4 | 100.0 [96.3–100.0] | 100.0 [99.2–100.0] | 100.0 [99.6–100.0] | 1.9 [1.9–1.9] | 9.5 [9.3–9.7] | 19.0 [18.6–19.4] | 91.7 [91.2–92.2] | 87.9 [87.2–88.6] |
| | Method | $p^+$ | P@100 | P@500 | P@1000 | R@100 | R@500 | R@1000 | AUROC | AP |
| Label Errors | NoiseRank | 32.3 | 35.0 [26.4–44.8] | 28.4 [24.6–32.5] | 29.6 [26.9–32.5] | 0.7 [0.5–0.8] | 2.7 [2.3–3.0] | 5.5 [5.1–6.1] | 48.2 [47.2–49.1] | 31.1 [30.2–32.1] |
| | CLearning | 32.3 | 34.0 [25.5–43.7] | 34.2 [30.2–38.5] | 33.7 [30.8–36.7] | 0.6 [0.5–0.8] | 3.2 [2.8–3.6] | 6.3 [5.7–6.9] | 48.1 [47.1–49.0] | 31.7 [30.8–32.8] |
| | FINE | 32.3 | 19.0 [12.5–27.8] | 24.4 [20.8–28.4] | 25.2 [22.6–28.0] | 0.4 [0.2–0.5] | 2.3 [1.9–2.6] | 4.7 [4.2–5.2] | 46.7 [45.8–47.6] | 30.0 [29.2–31.0] |
| | BHN | 32.3 | 25.0 [17.6–34.3] | 32.2 [28.3–36.4] | 32.4 [29.6–35.4] | 0.5 [0.3–0.6] | 3.0 [2.6–3.4] | 6.1 [5.6–6.6] | 54.7 [53.7–55.6] | 34.1 [33.1–35.2] |
| | SelfClean | 32.3 | 26.0 [18.4–35.4] | 39.8 [35.6–44.2] | 45.7 [42.6–48.8] | 0.5 [0.3–0.7] | 3.7 [3.3–4.2] | 8.6 [8.0–9.1] | 58.3 [57.3–59.2] | 38.7 [37.6–39.9] |

