# OpenReview forum: "CleanPatrick: A Benchmark for Image Data Cleaning"
_DMLR — Accepted by DMLR_

### Review · Reviewer_eBrV · 2025-12-22

**Recommendation:** 5
**Confidence:** 2

**Summary Of Contributions:**

This submission introduces CleanPatrick, a large-scale benchmark for image data cleaning grounded in a real-world medical imaging dataset (Fitzpatrick17k). The benchmark targets three practical data quality issues: off-topic samples, near-duplicate images, and label errors. The authors contribute (i) a substantial annotation effort involving hundreds of screened annotators and nearly half a million judgments, (ii) a probabilistic aggregation strategy inspired by item response theory to account for annotator reliability and task difficulty, and (iii) a standardised ranking-based evaluation protocol reflecting realistic human-in-the-loop data auditing workflows. The benchmark is accompanied by an empirical study comparing classical anomaly detection, perceptual hashing, self-supervised representation learning, and label-noise detection methods across the three issue types. The work aims to support more rigorous, data-centric evaluation of image cleaning methods beyond synthetic noise settings.

**Strengths:**

The paper makes a timely and relevant contribution to the growing data-centric ML literature by shifting attention from model architectures to dataset quality and auditing. The benchmark is grounded in a real, high-stakes application domain, which enhances its practical significance and discourages overfitting to artificial noise assumptions. The annotation effort is substantial and carefully designed, and the decision to evaluate cleaning methods via ranking metrics (AUROC, AP, precision@k) is well motivated by real-world workflows.

The empirical study is informative and highlights important gaps in current methods, particularly for label supervision issues. The paper is generally well written, structured, and transparent about several limitations. From a community perspective, the benchmark is likely to be useful for researchers working on data quality, dataset auditing, and weak supervision, especially in vision and medical imaging.

**Audience:**

Yes

**Broader Impact Concerns:**

The paper uses medical imagery, which raises standard concerns regarding dataset bias, annotator expertise, and downstream misuse. While the benchmark is positioned for auditing rather than diagnosis, clearer guidance on appropriate and inappropriate use cases would be beneficial. In particular, it should be emphasised that “label error” annotations do not constitute medical correction or clinical ground truth and should not be used to override expert diagnoses without additional evidence.

**Claims And Evidence:**

The empirical claims are generally supported by the presented experiments, particularly for off-topic detection and near-duplicate identification. However, claims about the intrinsic difficulty of label-error detection are not fully supported, as the evaluated methods are not designed for the benchmark’s conservative error definition. With appropriate reframing, the evidence would be convincing.

**Datasets And Benchmarks:**

The paper provides substantial detail on annotation procedures, aggregation, and evaluation protocols. With the requested clarifications on task semantics and representation dependence, the benchmark documentation would be sufficient to support reproducibility and responsible use.

**Extended Submissions:**

This submission does not appear to be an extension of a previously published work, or at least no such dependency is stated.

**Limitations:**

The benchmark’s notion of ground truth is epistemically limited by design: label errors are defined conservatively and based on visual plausibility rather than verified clinical truth. Near-duplicate labels depend on an embedding-based mining strategy that may miss certain duplicate types. Expert validation is restricted in scale, and some results are sensitive to annotation thresholds. As a result, conclusions must be interpreted as conditional on the benchmark’s specific definitions rather than as universal statements about data cleaning difficulty.

**Requested Changes:**

* Clarify and formalise task definitions, especially label errors. The current definition corresponds to “clearly wrong labels under conservative human judgment,” not general label noise or ground-truth incorrectness. This distinction must be explicit throughout the paper.

* Reframe claims regarding label-error detection. Poor performance should not be presented as evidence that label-error detection is broadly unsolved, but rather that existing methods are misaligned with the benchmark’s semantics.

* Explicitly acknowledge representation dependence in near-duplicate ground truth construction, and limit generalisation claims accordingly.

**Minor:**

* Add a threshold sensitivity analysis for expert-calibrated labels, particularly for label errors.

* Expand discussion on how different categories of label noise (ambiguous, hard, minority-class) are intentionally excluded and what implications this has.

* Improve clarity on the relationship between this benchmark and standard weak-supervision or noisy-label learning settings.

**Strengths And Weaknesses:**

**Strengths:**

Addresses an important and underexplored problem in data-centric ML: evaluating image data cleaning methods under realistic conditions.

Introduces a large-scale, human-annotated benchmark derived from a real medical dataset rather than synthetic corruptions.

Appropriately frames data cleaning as a ranking and prioritisation task, aligning evaluation with practical audit budgets.

Uses a non-trivial aggregation model that attempts to account for annotator ability and item difficulty instead of relying on majority vote.

Empirical results reveal meaningful distinctions between issue types, particularly the relative maturity of duplicate detection versus the difficulty of identifying label errors.

**Weaknesses:**

Core task definitions, especially “label error,” are underspecified and conflated with annotation procedure, leading to ambiguity about what is actually being evaluated.

Ground truth for near-duplicates is procedurally entangled with representation-based mining, limiting representation-agnostic claims.

Conclusions about the difficulty of label-error detection are stronger than what the benchmark semantics strictly support.

Expert calibration is limited in scale, and threshold sensitivity is not sufficiently analyzed.

Some baseline methods evaluated for label noise are misaligned with the benchmark’s conservative definition of error, complicating interpretation.